# Association of a Number of Tumor Foci with Ultrasound Patterns and Reoperation Rate in Well-Differentiated Thyroid Cancer (WDTC)—A Single-Center Analysis

**DOI:** 10.3390/cancers17223595

**Published:** 2025-11-07

**Authors:** Michał Miciak, Krzysztof Jurkiewicz, Natalia Kalka, Maja Reiner, Jakub Bogda, Szymon Biernat, Dorota Diakowska, Beata Wojtczak, Krzysztof Kaliszewski

**Affiliations:** 1Department of General Surgery, University Centre of General and Oncological Surgery, Faculty of Medicine, Wroclaw Medical University, 50-556 Wrocław, Poland; krzysztof.jurkiewicz@student.umw.edu.pl (K.J.); natalia.kalka@student.umw.edu.pl (N.K.); maja.reiner@student.umw.edu.pl (M.R.); jakub.bogda@student.umw.edu.pl (J.B.); szymon.biernat@student.umw.edu.pl (S.B.); 2Doctoral School, Wroclaw Medical University, 50-345 Wrocław, Poland; 3Division of Medical Biology, Faculty of Nursing and Midwifery, Wroclaw Medical University, 50-368 Wrocław, Poland; dorota.diakowska@umw.edu.pl; 4Department of Endocrine Surgery, University Centre of General and Oncological Surgery, Faculty of Medicine, Wroclaw Medical University, 50-556 Wrocław, Poland; beata.wojtczak@umw.edu.pl

**Keywords:** well-differentiated thyroid cancer, number of tumor foci, ultrasound imaging, thyroid surgery, reoperation rate, cancer management

## Abstract

Thyroid cancer (TC) is one of the most common endocrine malignancies, and accurate preoperative assessment is crucial for its optimal management. Ultrasound is the primary tool for diagnostic imaging, with features such as hypoechogenicity, microcalcifications, high vascularity, and irregular shape or margins often raising suspicion of malignancy. Another factor of interest is the number of tumor foci, which may influence prognosis and the choice of surgical strategy. In this study, we retrospectively analyzed 665 well-differentiated TC (WDTC) patients operated on between 2008 and 2024. We investigated relationships between the number of tumor foci, ultrasound patterns, and the reoperation rate. Our results showed a statistical association between the number of foci and suspicious ultrasound features, but the relationship was not linear. Importantly, the number of foci did not correlate with reoperation rates. These findings suggest that the number of tumor foci may provide limited additional value for TC diagnostics.

## 1. Introduction

Thyroid nodules are the most common disorders of the human endocrine system, with their prevalence estimated at approximately 2–6% in the general population based on large population studies [1]. Their occurrence is multifactorial and often correlated with age. Thyroid nodules are increasingly diagnosed in asymptomatic patients due to the widespread use of imaging studies. As a result, the incidence of thyroid cancer (TC) is also rising, largely due to the overdiagnosis of low-risk nodules detected incidentally [2]. TC is described as the ninth most common cancer worldwide, with the number of cases exceeding 580,000 in 2020. The highest incidence is observed in young women, among whom TC ranks as the fifth most frequently diagnosed cancer. Over the past three decades, its incidence has shown the greatest increase among all cancers; however, the mortality rate has remained consistently low [3]. Nearly 90% of TCs are classified as well-differentiated thyroid cancers (WDTCs), which include follicular and papillary subtypes. Within this group, the five-year cancer-specific survival rate is very high, approaching nearly 100% [4]. WDTC is typically asymptomatic in most patients; however, symptoms may occur if the tumor infiltrates adjacent neurovascular or muscular structures or metastasizes to lymph nodes. These symptoms can include a palpable neck mass, swelling, pain, movement tension, difficulty swallowing, hoarseness, or vocal changes. More concerning symptoms include hemoptysis or breathing difficulties caused by airway obstruction [5].

In ultrasound diagnostics, suspicion of malignancy in a thyroid nodule may be raised by patterns such as hypoechogenicity, microcalcifications, increased vascularity (marked intranodular flow), and irregular shape or margins, as well as a nodule size exceeding 1 cm. The Thyroid Imaging Reporting and Data System may be consulted, as it categorizes lesions based on ultrasound patterns and assesses their risk of malignancy. Subsequently, at the stage of fine-needle aspiration biopsy (FNAB), the nodule is categorized according to The Bethesda System for Reporting Thyroid Cytopathology. Categories V (suspicious for malignancy) and VI (malignant) usually qualify the patient for surgical treatment. The definitive diagnosis of TC is established based on postoperative histopathological examination, in accordance with the TNM Classification for Thyroid Cancer and the WHO Classification of Thyroid Neoplasms [6,7,8,9,10].

Determining pathological features such as bilaterality, multifocality, the number of tumor foci, vascular and capsular invasion, extrathyroidal extension, positive surgical margins, and the presence of lymph node metastases is also highly important in TC management. Together with ultrasound patterns, these features can be used to determine the extent of thyroidectomy (hemi- or total), decide on the need for diagnostic or therapeutic cervical lymphadenectomy, assess the need for radical surgery according to the reoperation risk, and guide the frequency of follow-up visits [11]. Multifocality, bilaterality, or extrathyroidal extension are examples of features with a well-defined impact on predicting nodule malignancy and TC advancement. However, the specific number of tumor foci (1, 2, 3, or ≥4) represents an additional parameter that can significantly influence TC diagnostics, management, and predictability [12,13,14].

Thus, the aim of this study is to determine the association between the number of tumor foci and the presence of unfavorable ultrasound patterns and to assess whether a higher number of foci is significantly associated with the reoperation rate, with additional stratification by TNM stage and sex. This research attempts to find out the clinical value of subsequent diagnostic parameters and further contributes to the field of TC management.

## 2. Materials and Methods

We retrospectively analyzed the medical records of 6244 patients treated at the University Centre of General and Oncological Surgery, Department of General Surgery, Wroclaw Medical University. The timeframe of the data collected for this study extended from January 2008 to December 2024. Patients were treated surgically with intraoperative nerve monitoring for nodular thyroid diseases, including benign tumors, malignant tumors, and thyroid goiter. The process of selecting the study group is shown in Figure 1.

Patient data were retrospectively analyzed in a fully anonymized dataset compiled using medical records. Verbal consent for participation was obtained and documented for all patients. The authors had no access to identifiable patient information and did not have direct contact with the study participants. This study’s protocol was approved by the Bioethics Committee of Wroclaw Medical University, Poland (Approval No. KB-241/2023).

A group of 665 patients with confirmed WDTC was further analyzed. The majority of patients presented with papillary TC, including 629 cases (86.9%), and 36 cases (5.0%) of follicular TC, accounting for nearly 92% of all TCs. The remaining histological subtypes included: medullary TC, anaplastic TC, lymphoma, squamous cell tumor, sarcoma, myeloma, and secondary lesions. However, these cases were not included in the study group, as they differ in management strategies and are associated with a less favorable prognosis.

All patients resided in geographically iodine-sufficient regions prior to the evaluation. Each patient underwent thyroid ultrasound and FNAB before surgery. Ultrasound patterns like hypoechogenicity, microcalcifications, high vascularity, or irregular tumor shape or margins were reviewed and interpreted by experienced radiologists. In the context of high vascularity, the presence of marked intranodular flow was defined as a suspicious pattern. Postoperative histopathological evaluation was performed by three independent pathologists, noting the number of tumor foci (1, 2, 3, or ≥4), multifocality, bilaterality, extrathyroidal extension, and pTNM stage.

Patients underwent primary hemi-thyroidectomy or total thyroidectomy where indicated, in addition to diagnostic lymphadenectomy of the mid-cervical compartment. Prophylactic lymphadenectomy was not performed routinely. Following surgical treatment, follow-up evaluations were performed at approximately 3–4-month intervals during the first five years. These assessments included ultrasound imaging and laboratory testing of serum TSH and thyroglobulin levels. Thereafter, follow-up was conducted annually, or more frequently in cases where symptoms suggestive of TC recurrence were observed, with potential qualification for adjuvant management, including reoperation. Indications for reoperation included mainly clinical nodular recurrence in the contralateral thyroid lobe and completion thyroidectomy following TC diagnosis after an initial hemi-thyroidectomy, in cases where suspicious features (e.g., multifocality, vascular or capsular invasion, extrathyroidal extension) were identified on histopathological examination. Also, the additional radicalized surgical intervention with lymphadenectomy of additional cervical compartments after initial total thyroidectomy was intermittently indicated in multifocal cases where postoperative imaging studies (CT scintigraphy/SPECT) were positive during follow-up. Finally, the reoperation rate was determined.

### Statistical Analysis

Qualitative data were expressed as counts and percentages, while quantitative data were presented as medians with interquartile ranges, reflecting the non-normal distribution of data. The Shapiro–Wilk test was employed to evaluate the normality of quantitative variables. For the purposes of certain analyses, the features studied were also encoded in binary form. Group differences were examined using the two-tailed Mann–Whitney U test for continuous variables in two groups. For analyses involving more than two groups, the Kruskal–Wallis test was used. The chi-square test was applied to compare categorical variables. For additional analyses stratified by TNM stage and sex, 95% Wilson confidence intervals were calculated. Group comparisons were performed using the chi-square or Fisher’s exact test, as appropriate. A logistic regression model including interaction terms was applied to formally evaluate effect modification. All statistical analyses were conducted using Statistica version 10.0 (StatSoft Inc., Tulsa, OK, USA). A *p*-Value of less than 0.05 (<0.05) indicated a statistically significant difference between the variables.

## 3. Results

### 3.1. Overview of the Study Group

The demographics and clinical characteristics of the study group are presented in Table 1. The majority of patients were female (85.0%) and younger than 55 years (57.9%), which was considered young. Most patients underwent total thyroidectomy and did not require reoperation (73.1% and 77.6%, respectively). A positive FNAB result (Bethesda category V or VI) was obtained in 97.7% of patients. Most TCs were unilateral and solitary, presenting as a single nodule, and were classified as stage I according to the TNM system, commonly as pT1aN0M0. Certain nodal and metastatic stages (reported as pNx and pMx) could not be determined due to the absence of routine prophylactic lymphadenectomy.

### 3.2. Number of Foci Status and Ultrasound Pattern Distribution in WDTC

The data are presented in Figure 2. The suspicious ultrasound patterns analyzed included hypoechogenicity, microcalcifications, high vascularity, and irregular tumor shape or margins. Most patients (80%) presented with hypoechogenicity. Microcalcifications were identified in 53.1% of patients, while high vascularity and an irregular tumor shape or margins were observed less frequently (48.1% and 48.3%, respectively). With regard to the number of tumor foci, the majority of patients had a single TC focus (80.3%), whereas the smallest subgroup comprised patients with ≥4 TC foci (1.2%).

### 3.3. Analysis of the Association Between WDTC Number of Foci and Ultrasound Patterns

We analyzed whether an increasing number of tumor foci (1, 2, 3, or ≥4) was associated with a higher number of studied ultrasound patterns. Table 2 presents the mean number of patterns (±SD) for each category of number of tumor foci, as shown in Figure 3. The lowest mean number of ultrasound patterns was observed in patients with 2 foci, after which the values increased again in the groups with 3 and ≥4 foci. Differences between groups were statistically significant (Kruskal–Wallis test, *p* < 0.05). Therefore, a simple linear increase in the number of negative ultrasound patterns with a rising number of tumor foci was not observed. This may be related to the unequal distribution of patients across groups, as the one-focus category was statistically the largest. Thus, while a statistically significant association exists between the number of tumor foci and the number of negative ultrasound features, the observed trend is not linear.

Figure 4 presents the distribution of ultrasound pattern groups (0, 1, 2, and more) according to the number of tumor foci (1, 2, 3, or ≥4). It can be observed that in each focus category, the subgroup with the greatest number of ultrasound patterns was consistently predominant.

On that basis, the patients with the full set of all four ultrasound patterns (*n* = 48) were compared with the remaining patients (*n* = 617). The median number of tumor foci in both groups was 1.0, and the mean (±SD) was approximately 1.25 (±0.73 in the group with all patterns present versus ±0.58 in the remaining group). The Mann–Whitney U test revealed no significant difference (*p* = 0.273). Thus, the patients with the complete set of ultrasound patterns did not have a statistically higher number of foci compared to the others. Similarly, due to the limited number of patients without any pattern (only six cases), the subgroup with at least one pattern and the subgroup without any were not analyzed separately.

### 3.4. Analysis of the Association Between WDTC Number of Foci and Reoperation Rate

In the WDTC group analyzed, 149 of 665 patients (22.4%) required reoperation. Table 3 presents the reoperation rates according to the number of tumor foci. The highest reoperation rate was observed in patients with 3 foci (25.9%), while the lowest was in those with ≥4 foci (0%); however, this group was statistically the smallest in size. The chi-square test showed no significant difference between these groups (χ^2^ = 2.538; df = 3; *p* = 0.469). Therefore, no significant trend was observed in reoperation frequency with increasing numbers of tumor foci.

In relation to clinical and demographic factors, an analysis of the association between the number of WDTC foci and the reoperation rate was additionally performed, stratified by TNM stage and sex. Across all TNM stages (I, II, III and IV), neither the chi-square nor Fisher’s exact test demonstrated statistically significant differences (*p* = 0.115–1.00). For stage IV, the sample size (*n* = 14) was too limited for reliable statistical conclusions. Thus, TNM stage did not appear to affect the reoperation rate within the respective groups defined by the number of tumor foci (1, 2, 3, or ≥4). For the analysis stratified by sex, the distribution of reoperation rates across the categories of tumor foci (1, 2, 3, or ≥4) also did not differ significantly in either males or females (chi-square or Fisher’s exact test; all *p* > 0.05). These analyses indicate that sex did not have a significant influence on the association between the number of foci and reoperation rate in our study cohort. The final fitted logistic regression model, which included interaction terms (number of foci × TNM stage; number of foci × sex), yielded an odds ratio of 0.90 (95% CI: 0.65–1.25; *p* = 0.53), indicating no significant interaction between the number of tumor foci and the reoperation rate across TNM stage and sex stratifications.

## 4. Discussion

This study aimed to examine the association between the number of tumor foci and specific ultrasound patterns, including hypoechogenicity, microcalcifications, high vascularity, and irregular tumor shape or margins. Furthermore, we assessed the impact of the number of foci on the frequency of WDTC reoperations.

### 4.1. Number of Foci Distribution

In terms of tumor foci distribution, solitary lesions predominated in our study (534; 80.3%), followed by cases with two foci (96; 14.4%) and three foci (27; 4.1%). The presence of four or more foci was the least common and was observed in only eight patients (1.2%). The literature demonstrates considerable variation in the reported distribution of tumor foci. In a study by Kwon H. et al. [15], the proportions of cases with 1, 2, 3, and ≥4 foci were 61.1%, 25.0%, 8.9%, and 5.0%, respectively. Similarly, Gao Y. et al. [16] reported 54.6%, 26.8%, 12.3%, and 6.3% for the same categories. When viewed in the broader context of multifocality, the literature-reported prevalence of approximately 30% among patients with papillary TC aligns well with our observed rate of 26.9% [15,16]. Studies using alternative grouping approaches revealed comparable trends. Evans Harding N. et al. [14] identified 65% unifocal and 35% multifocal (2 to ≥5) TCs; Elbasan O. et al. [17] reported 69.3% of TCs with one focus, 18% with two foci, and 12.7% with three or more foci; while Li D. et al. [18] found 51.5% TCs with a single focus, 46.1% with two to four foci, and 2.4% with five or more foci. In addition to the number of foci, another commonly reported parameter used for grouping TCs is the tumor size, with a distinction between microcarcinomas and macrocarcinomas [14,17,18]. A clear predominance of solitary lesions and a progressively smaller proportion of groups with higher numbers of tumor foci can be observed, consistent with the findings of our study.

### 4.2. Association with Studied Ultrasound Patterns

Regarding ultrasound patterns, the observed prevalence in our cohort was 80%, 53.1%, 48.1%, and 48.3% for hypoechogenicity, microcalcifications, high vascularity, and irregular tumor shape or margins, respectively. These results are consistent with previously published data. Reported rates of hypoechogenicity range from 71% to 91.7%, depending on the classification of the parameter (marked, moderate, or mild) and the anatomical location of the thyroid lesion (lobe or isthmus). The authors report that markedly or moderately hypoechoic nodules represent the highest malignancy risk category and note that suspicious ultrasound patterns tend to predominate in nodules arising from the thyroid isthmus [19,20]. The prevalence of microcalcifications has been reported to be between 33.7% and 59.3% and is influenced by additional factors such as hyperthyroidism or the presence of coronary artery calcifications. For this parameter, discrepancies also can be observed across the literature, with authors dividing it into various subcategories such as no-calcification, microcalcifications, macrocalcifications, mixed calcifications, rim calcifications, or isolated calcifications, depending on the adopted classification criteria [21,22]. Regarding the prevalence of high vascularity, the literature shows a broad range from 16.7% to 91.7%, likely due to variability in ultrasound criteria. However, increased central vascularity within a nodule is generally considered the most diagnostically relevant feature of TC. Meta-analyses estimating this pattern in approximately half of TC cases align with our findings; however, the parameter of vascularity should be considered as auxiliary to the TIRADS score [10,23,24,25]. The reported prevalence of irregular tumor shape varies from 39.1% to 46.7% [26,27]. This feature is sometimes described interchangeably with irregular or spiculated margins, which may introduce interpretive variability. Nevertheless, the analyzed ultrasound pattern of irregular tumor shape or margins remains predictive of thyroid malignancy and is adopted in the TIRADS system.

Our analysis revealed a statistically significant association between the number of tumor foci and the mean number of ultrasound patterns (Kruskal–Wallis test, *p* < 0.05). However, this relationship was not linear, and patients presenting with the complete set of ultrasound patterns did not exhibit a statistically higher number of foci compared to others (Mann–Whitney U test, *p* = 0.273). This indicates that a greater number of suspicious ultrasound patterns does not necessarily correspond to a higher number of tumor foci upon histopathological evaluation. To our knowledge, no previous studies have specifically examined the association between ultrasound patterns and the number of tumor foci, although some analyses have explored similar relationships using multifocality as a binary variable. Studies indicate that hypoechogenicity, microcalcifications, high vascularity, and irregular tumor shape or margins can be associated with multifocal TC. Conversely, multifocality or even bilaterality may sometimes remain undetected during pre-surgical evaluations, which could introduce diagnostic bias when compared with determined ultrasound patterns [28,29,30,31]. However, the discussed association appears to be indirect, and the specific influence of the number of tumor foci cannot be clearly determined without further large-cohort studies.

### 4.3. Association with Reoperation Rate

The overall reoperation rate in the study cohort was 22.4% (149 out of 665 WDTC cases), irrespective of the number of tumor foci. Previous reports describe a rate of reoperation of approximately 20%, although up to 30% of patients with TC who undergo initial treatment may experience recurrence. As previously mentioned, this may be attributable to the inherent limitations in reliably determining multifocality during preoperative evaluation. Our findings are consistent with these data. Moreover, recent trends indicate a decreasing rate of reoperations following hemi-thyroidectomy (from 33.9% to 14.2%) and a persistently low rate after total thyroidectomy (around 0.2%), as reported in large national cohort studies [32,33,34].

Subsequent analysis did not reveal a statistically significant association between the number of tumor foci and the reoperation rate (chi-square test, *p* = 0.469), indicating that the frequency of reoperation did not increase with a higher number of TC foci. The literature varies in this area. Among studies investigating recurrence, Qu N. et al. [35] reported that recurrence and survival parameters decreased significantly with an increasing number of tumor foci (*p* = 0.041), particularly in cases with three or more foci. Similarly, Evans Harding N. et al. [14] found that the presence of four or more foci was associated with poorer outcomes, suggesting that this threshold may serve as a potential cutoff for treatment intensification. Kim H. et al. [36] also demonstrated a correlation between recurrence and the number of foci, with hazard ratios of 1.45 and 1.95 for patients with two or three or more foci, respectively. Conversely, Kwon H. et al. reported that the risk of recurrence was comparable between unifocal and bifocal cases [14,15,35,36]. Our analysis stratified by TNM stage and sex did not demonstrate any significant effect of these factors (*p* > 0.05) on the reoperation rate across the different numbers of tumor foci. In the literature, there can be found references to the multifocality parameter. A higher TNM stage is often reported as a factor associated with tumor recurrence and, indirectly, with the likelihood of reoperation. However, such findings typically concern more advanced nodal metastases (N1b), which in our cohort accounted for only 1.9% of cases. Regarding sex, the literature presents divergent conclusions. Depending on the study, male sex is reported either as not associated with the risk of recurrence or as not being an independent prognostic factor. However, other authors emphasize that male sex is associated with a higher risk of tumor recurrence in univariate analysis. Nevertheless, these studies do not always specifically address multifocality and they highlight another relevant aspect in this context, that the likelihood of recurrence in TCs harboring mutations such as BRAF^V600E^ may vary according to sex [37,38,39,40,41]. Several studies have reported that a higher number of tumor foci and multifocality correlate with histopathological features of TC advancement, including extrathyroidal extension, capsular and vascular invasion, and lymph node metastases. These associations may indirectly contribute to increased recurrence risk and the reoperation rate. Although cutoff thresholds vary among studies, the presence of ≥3 or ≥4 foci is most commonly identified as clinically significant [13,35,42,43,44,45,46].

In a broader context, in addition to the number of foci, multifocality itself is commonly evaluated as a predictor of recurrence and reoperation. However, conclusions remain inconsistent across studies. Woo J. et al. [40] identified multifocality as an independent predictor of recurrence, suggesting the need for more aggressive treatment strategies and follow-up. Similarly, Wu Z. et al. [47] reported that patients with bilateral multifocal TC exhibited the highest recurrence rates compared with other subgroups. These findings are consistent with the conclusions of the previously cited meta-analysis by Kim H. et al. [36,40,47]. In contrast, Wang F. et al. [41] reported a recurrence rate of 19.8% in multifocal TCs, concluding that multifocality was not an independent prognostic risk factor. Other studies similarly indicate that this parameter does not improve the predictability of recurrence, and in some cases, multifocality does not appear to confer worse outcomes in WDTC. Nevertheless, multifocality is often perceived as a useful parameter for guiding intensified TC management, particularly in light of reports indicating an increasing trend in the number of tumor foci in recent years [14,15,41,48].

### 4.4. Study Limitations and Future Directions

Our study had several limitations. First, it was conducted at a single surgical center, which may limit the generalizability of the findings compared to multicenter studies. Second, the retrospective nature of this study inherently introduces certain inaccuracies, as diagnostic and management guidelines for TC have evolved over time. For instance, subsequent revisions of the TIRADS classification introduced updated ultrasound patterns and corresponding management recommendations for each category (recommendations of ultrasound surveillance or FNAB), while current guidelines now permit active surveillance in selected cases of thyroid microcarcinoma (<1 cm in size). Third, this study did not incorporate molecular or genetic testing data for TC, primarily due to the broad temporal scope of the study and institutional limitations. Finally, the core data analyzed were histopathological and ultrasound findings. This may have introduced selection bias, as only patients with histologically confirmed TCs were included in the final analysis.

In future studies related to these findings, broader prospective research could investigate the number of tumor foci in relation to other histopathological features of TC advancement combined with ultrasound patterns. Evaluating the coexistence of the number of foci with established prognostic parameters such as extrathyroidal extension, capsular invasion, and lymph node metastases may help define a potential cutoff point for clinical relevance. In the current era of molecular medicine, it would be reasonable to incorporate molecular testing results alongside ultrasound and histopathological data. Such an approach may contribute to a more accurate selection of methods in the context of personalized medicine. Expanding the analysis to include other potentially relevant thyroid imaging techniques is also worth considering.

## 5. Conclusions

Our findings highlight the association between the number of WDTC foci and suspicious ultrasound patterns such as hypoechogenicity, microcalcifications, high vascularity, and irregular tumor shape or margins. However, the lack of a consistent correlation between ultrasound patterns and the increasing number of foci, together with the absence of an impact on reoperation rates, suggests that this parameter, when considered in isolation, should not guide clinical decisions for more intensive follow-up or the extent of reoperation. The number of tumor foci should be interpreted in the broader context of TC multifocality in diagnostics and management.

## Figures and Tables

**Figure 1 cancers-17-03595-f001:**
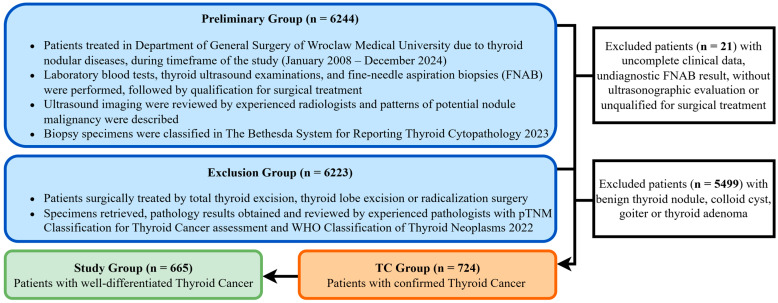
Study group selection process. A total of 6244 medical records of patients admitted to the study center during the study period were evaluated. The patients underwent a diagnostic process that included ultrasound examination, fine-needle aspiration biopsy, surgical treatment for thyroid cancer, and postoperative pathological examination. A total of 665 records that met the criteria were selected as a study group for further analysis.

**Figure 2 cancers-17-03595-f002:**
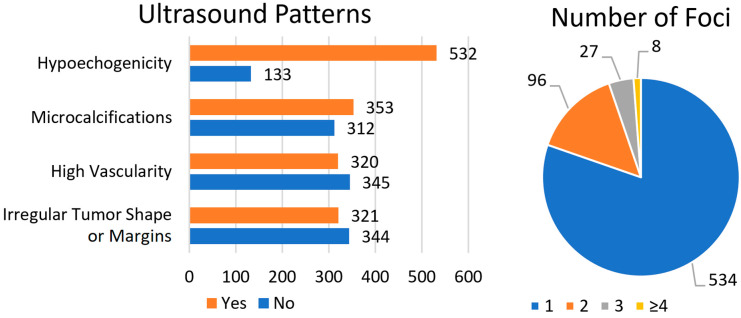
Distribution of studied ultrasound patterns: hypoechogenicity, microcalcifications, high vascularity and irregular tumor shape or margins (**left side**) and number of tumor foci (**right side**) across the WDTC study group.

**Figure 3 cancers-17-03595-f003:**
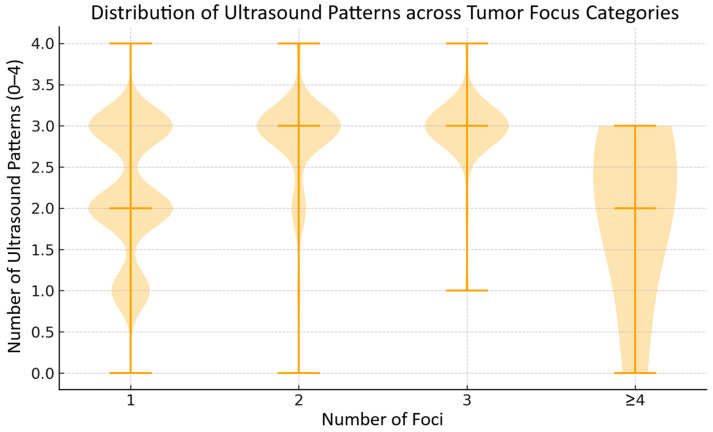
Distribution of mean number of studied ultrasound patterns: hypoechogenicity, microcalcifications, high vascularity and irregular tumor shape or margins (vertical axis) according to number of tumor foci groups: 1, 2, 3, and ≥4 (horizontal axis). Note: Horizontal lines indicate the median values, while bolded segments represent the quartiles.

**Figure 4 cancers-17-03595-f004:**
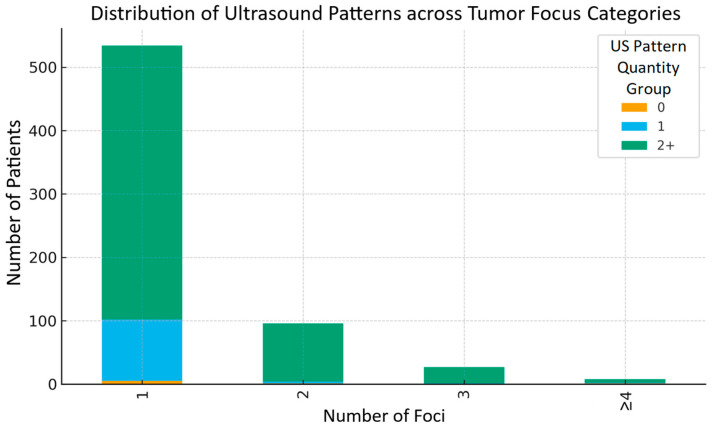
Distribution of ultrasound patterns quantity groups: 0, 1, 2 and more (color bars), presented as quantity of cases (vertical axis) according to number of tumor foci groups: 1, 2, 3, and ≥4 (horizontal axis).

**Table 1 cancers-17-03595-t001:** Demographics and clinical characteristics of the WDTC patient study group. A *p*-Value of < 0.05 represents a statistically significant difference.

Feature		Quantity (Percentage)	*p*-Value
Sex	Male	100 (15.0%)	0.11
	Female	565 (85.0%)	
Age	<55 years	385 (57.9%)	<0.05
	≥55 years	280 (42.1%)	
Type of Surgery	Total thyroidectomy	486 (73.1%)	<0.05
	Hemi-thyroidectomy	179 (26.3%)	
Need for Reoperation	Yes	149 (22.4%)	<0.05
	No	516 (77.6%)	
FNAB Result	Positive (Bethesda V or VI)	650 (97.7%)	<0.05
	Negative (Bethesda II, III, or IV)	15 (2.3%)	
Bilaterality	Yes No	62 (9.9%) 603 (90.1%)	<0.05
Focality	Solitary Multifocal	486 (73.1%) 179 (26.9%)	<0.05
Histological subtype of WDTC	Papillary TC	629 (94.6%)	0.07
	Follicular TC	36 (5.4%)	
TNM Stage	I	521 (78.3%)	<0.05
	II	106 (15.9%)	
	III	24 (3.6%)	
	IV	14 (2.2%)	
pT Stage	pT1a	285 (42.9%)	<0.05
	pT1b	273 (41.1%)	
	pT2	79 (11.9%)	
	pT3	17 (2.5%)	
	pT4a	3 (0.4%)	
	pT4b	8 (1.2%)	
pN Stage	pN0	439 (66.0%)	<0.05
	pN1a	182 (27.4%)	
	pN1b	13 (1.9%)	
	pNx	31 (4.7%)	
pM Stage	pM0	546 (82.1%)	<0.05
	pM1	25 (3.8%)	
	pMx	94 (14.1%)	

Annotation. FNAB—fine-needle aspiration biopsy; WDTC—well-differentiated thyroid cancer.

**Table 2 cancers-17-03595-t002:** Mean (±SD) number of negative ultrasound features according to the number of tumor foci. Differences between groups are statistically significant (*p* < 0.05).

Number of Foci	Quantity	Mean (±SD) Number of Ultrasound Patterns
1	534	2.36 ± 0.80
2	96	1.96 ± 0.56
3	27	2.22 ± 0.58
≥4	8	2.50 ± 1.20

**Table 3 cancers-17-03595-t003:** Reoperation rates according to the number of tumor foci. Differences between groups are not statistically significant (*p* = 0.469).

Number of Foci	Quantity	Reoperation Quantity	Reoperation Rate
1	534	121	22.7%
2	96	21	21.9%
3	27	7	25.9%
≥4	8	0	0%

## Data Availability

The datasets used and/or analyzed during this research are available from the corresponding author upon reasonable request.

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
