# Peer review of "Association of a Number of Tumor Foci with Ultrasound Patterns and Reoperation Rate in Well-Differentiated Thyroid Cancer (WDTC)—A Single-Center Analysis"

_cancers, 2025, doi:10.3390/cancers17223595_

Round 1
Reviewer 1 Report
Comments and Suggestions for Authors
The issue of multifocality, especially in papillary carcinoma, is a highly significant clinical problem. Therefore, the research undertaken by the authors is justified. Before publication, please provide clarification and answers to the following questions:
-
What was the reason for reoperation, particularly in the group with three confirmed carcinoma foci?
-
Did the assessment of lesion margin irregularity also include the presence of ill-define border?
-
Did the authors consider the complete absence of vascular flow in papillary microcarcinomas as a characteristic pattern of malignant lesions?
Reviewer 2 Report
Comments and Suggestions for Authors
-
Provide subgroup analysis stratified by TNM stage for reoperation rate analysis
-
Clarify temporal variations and consider sensitivity analysis excluding the earliest study period
-
Better characterize follow-up methods and duration
-
Provide power calculations for the negative findings regarding reoperation rates
-
More thoroughly discuss the discrepancies with existing literature
-
Consider excluding non-WDTC histologies from the primary analysis rather than analyzing separately.
-
Provide sex-stratified analyses
-
Improve figure legends for clarity
-
Consider reframing conclusions to emphasize clinical implications
Minor language editing is required.
Round 2
Reviewer 1 Report
Comments and Suggestions for Authors
I confirm my acceptance of this version of the manuscript.